

# Combinative effects of *β*-elemene and propranolol on the proliferation, migration, and angiogenesis of hemangioma

Zhenyu Wang[1,*], Yinxian Chen[1,*], Lin Yang[2], Dunbiao Yao[3] and Yang Shen[1]

[1] Department of Pediatric Orthopedics, Shanghai Children's Hospital, School of Medicine, Shanghai Jiaotong University, Shanghai, China
[2] Department of Urinary Surgery, Cengong County People's Hospital, Guizhou, China
[3] Department of Orthopedics, Cengong County People's Hospital, Guizhou, China
[*] These authors contributed equally to this work.

## ABSTRACT

Hemangioma (HA) is one of the most common benign vascular tumors among children. Propranolol is used as the first-line treatment for hemangioma and is a non-selective blocker of the *β*-adrenergic receptor. *β*-elemene is a compound extracted from Rhizoma zedoariae and has been approved for the treatment of tumors in clinical practice. However, the combinatorial effects of *β*-elemene and propranolol in the treatment of HA remains unclear. This study explored the combinative effects and mechanisms of *β*-elemene and propranolol using hemangioma-derived endothelial cells (HemECs). Cytotoxic assays showed that the combinatorial treatment of *β*-elemene and propranolol did not increase the cytotoxic effects of HemECs. Furthermore, functional analysis showed that the combinatorial treatment with *β*-elemene and propranolol significantly inhibited the proliferation, migration, and tube formation of the HemECs compared to the single treatment regimens. Mechanistic analysis showed that combinative treatment with *β*-elemene and propranolol synergistically down-regulated the hypoxia-inducible factor-1 alpha/vascular endothelial growth factor-A (HIF-1-$\alpha$/VEGFA) signaling pathway. Additionally, in a xenograft tumor model, angiogenesis in the combinatorial treatment group was significantly lower than in the control, propranolol, and *β*-elemene treatment alone groups. Our results suggest that *β*-elemene combined with propranolol can significantly inhibit the proliferation, migration, and tube formation of HemECs via synergistically down-regulating the HIF-1-$\alpha$/VEGFA signaling pathway without increasing any cytotoxic side effects.

# INTRODUCTION

Hemangioma (HA) is a type of benign vascular neoplasm characterized by the rapid growth of immature endothelial cells in the proliferative phase as well as the spontaneous regression of the neovasculature within the involutional phase (*Boscolo & Bischoff, 2009*; *Chen et al., 2022*). HA is usually harmless; however, 10–15% HA can cause ulcerative, disfiguring, and life-threatening complications (*Leaute-Labreze, Harper & Hoeger, 2017*).

Corresponding author
Yang Shen, doctor_sheny@163.com

Propranolol is one of the first non-selective $\beta$-adrenergic receptor inhibitors in clinical use, and has become the first-line treatment for hemangioma since the discovery of its inhibitory effect (*Leaute-Labreze et al., 2008*). A large international randomized clinical trial documented the efficacy and relative safety of propranolol in patients with HA (*Pope et al., 2022*). Previous studies have shown that propranolol inhibits the angiogenesis of normal endothelial cells, as well as the angiogenesis of hemangioma (*Chim et al., 2012*; *Lamy et al., 2010*). It has also been shown that propranolol inhibits the proliferation and induces the apoptosis of endothelial cells within hemangioma, which was mediated by the suppression of vascular endothelial growth factor A (VEGFA) expression (*Tavakoli et al., 2017*; *Wu et al., 2021b*).

However, the ability of propranolol to cross the blood–brain barrier with potential unproven long-term neurocognitive effects, as well as the higher pharmacokinetic variability of propranolol (*Agesen et al., 2019*), suggests that propranolol still produces adverse reactions in a clinical setting, such as respiratory, cardiac, metabolic disorders, sleep disturbance, behavioral changes, and effects on memory (*Droitcourt et al., 2018*). Therefore, the use of propranolol to treat hemangioma still has certain limitations within clinical treatment. In recent years, further improvements in the efficacy as well as reducing the side effects of propranolol by combining it with other therapeutic drugs has been an active area of significant hemangioma research (*Lou, Xu & Huo, 2018*; *Qiao et al., 2020*).

$\beta$-elemene (1-methyl-1-vinyl-2,4-diisopropenyl-cyclohexane) is an active compound purified from Rhizoma zedoariae (*Fang et al., 2018*; *Wu et al., 2022*). $\beta$-elemene plays an important role in co-biosynthesis, attenuating inflammation and neuropathic pain, inhibiting fibrosis and reversing the drug resistance of tumor cells. In recent years, more and more discoveries have suggested that elemene has a good anti-tumor effect, and $\beta$-elemene is the major active component with anti-cancer potential in various cancers (*Fu, Gao & Xing, 2022*). $\beta$-elemene has been reported to possess anti-tumor activity in various malignant tumors such as colorectal, lung, and breast cancer (*Cai et al., 2021*; *Chen et al., 2020*; *Han et al., 2021*). $\beta$-elemene also has a broad spectrum of antitumor activity, high efficiency, and low toxicity (*Kamran et al., 2022*; *Ni et al., 2022*). As a radiotherapy sensitizer, $\beta$-elemene increases the sensitivity of tumor radiotherapy and circumvents drug resistance (*Tong et al., 2020*), and has been recently formulated into a new drug and approved by the State Food and Drug Administration of China for the treatment of solid tumors. In our previous work we have shown that high concentrations of $\beta$-elemene inhibits hemangioma growth and angiogenesis (*Wang et al., 2021*). However, the combined effect of $\beta$-elemene and propranolol as well as its molecular actions on angiogenesis-dependent disorders has not been reported. Therefore, we carried out this study to systematically study the anti-hemangioma effects of a combinatorial therapeutic approach using $\beta$-elemene and propranolol.

In this study, our data suggest that $\beta$-elemene is a new therapeutic drug. In combination with propranolol $\beta$-elemene significantly inhibits the proliferation, migration, and tube formation of HemECs by synergistically down-regulating the HIF-1-$\alpha$/VEGFA signaling pathway without increasing cytotoxic side effects.

## MATERIALS & METHODS

### Cell culture and treatment

HemECs were purchased from the Institute of Otwo Biotech (Shenzhen, China), and human umbilical vein endothelial cells (HUVECs) were purchased from the American Type Culture Collection (ATCC; Manassas, VA, USA). Cells were cultured at 37 °C under 5% $CO_2$ and 95% air, HemECs were maintained in a DMEM medium supplemented with 10% FBS (Gibco, Billings, MT, USA) and 1% penicillin-streptomycin (FBS, Hyclone, Logan, UT, USA), HUVECs were maintained in ECM medium (Gibco, Billings, MT, USA). HemECs were exposed to either $\beta$-elemene (Dalian Holley Kingkong Pharmaceutical Co., Dalian, China) or propranolol (Sigma-Aldrich, Merck, St. Louis, MO, USA), or a combined treatment of $\beta$-elemene and propranolol for 24 h prior to analysis.

### Calcein-AM/PI double staining

HemECs were incubated in six-well plates at a density of $5 \times 10^4$ cells per well, Then the cells were treated with $\beta$-elemene or propranolol for 24 h. Next a Calcein-AM/PI Double Staining Kit (Cat No. MA0361; Meilun Biotech, Jiujiang, China) was used according to the manufacturer's instructions, then incubated at 37 °C for 15 min. An excitation wavelength of $490 \pm 10$ nm was used to image living and dead cells using a fluorescence microscope (Olympus, Tokyo, Japan).

### Cell counting Kit-8 (CCK-8) assay and LDH assays

Following the manufacturer's instructions, cell proliferation was tested *via* a CCK-8 kit (Cat No. C0042; Beyotime Institute of Biotechnology, Jiangsu, China). $1 \times 10^4$ cells per well were seeded into a 96-well plate for 24 h and treated with $\beta$-elemene or propranolol for 24 h, 48 h, and 72 h according to the study design. After the treatment, 10 µl of CCK-8 solution was added to each well and incubated for another 2 h at 37 °C. The optical cell density was measured at 450 nm using a microplate reader (Thermo Fisher Scientific, Waltham, MA, USA). Additionally, we collected cell culture supernatants and used LDH activity kits (Cat No. C0017; Beyotime Institute of Biotechnology, Jiangsu, China) to measure lactate dehydrogenase (LDH) levels.

### Scratch test

We plated $4 \times 10^4$ HemECs per well for the scratch test in six-well plates. After the HemECs reached 100% confluency, the wounds were made using a yellow pipette tip. The HemECs were treated with $\beta$-elemene and propranolol and measured after 24 h. The experiment was repeated in triplicate. Data were expressed as the mean $\pm$ SD.

### Tube formation assay

Matrigel Matrix (Corning, Corning, NY, USA) was added to each well of a 96-well plate and incubated at 37 °C for 30 min. Then we added HemECs ($2 \times 10^4$ cells/well) to 100 µL serum-free DMEM pipetted onto the Matrigel-coated plates and incubated at 37 °C for 8 h. The capillary-like structures were observed and imaged (Olympus, Toyko, Japan). The tube number was quantified using ImageJ software.

## Western blot

Cells were harvested by centrifugation and lysed using RIPA lysis buffer (Cat No. G 2002-100ML; Servicebio, Wuhan, China) containing 1% protease inhibitor (Cat No. G2008-1ML; Servicebio, Wuhan, China) after treatment with $\beta$-elemene or propranolol for 24 h. Antibodies for HIF-1-$\alpha$ (Cat No. sc-13515; Santa Cruz Biotechnology, Dallas, TX, USA), VEGFA (Cat No. ab214424; Abcam, Cambridge, UK), phospho-Akt (Cat No. T56569; Abmart, Shanghai, China), Akt (Cat No. M63147; Abmart, Shanghai, China), phospho-Erk (Cat No. T57165; Abmart, Shanghai, China), Erk (Cat No. M63378, Abmart, Shanghai, China) were used in western blotting analyses. $\beta$-Tubulin (Cat No. M20005; Abmart, Shanghai, China) was used to confirm equal protein loading.

## Total RNA extraction and real-time PCR

Total RNA was isolated with TRIzol Reagent (Thermo Fisher Scientific, Waltham, MA, USA) according to manufacturer's protocol, and next the quantity and quality of RNA were assessed with Nanodrop 2000 (Thermo Fisher Scientific, Waltham, MA, USA), then approximately 1 microg of total RNA was reverse transcribed *via* HiScript III RT SuperMix (+gDNA wiper) (Vazyme, Nanjing, China). The qRT-PCR was carried out using a AceQ Universal SYBR qPCR Master Mix (Vazyme, Nanjing, China) with conditions following the manufacturer's instructions. Relative gene expression levels were calculated using the $2^{-\Delta\Delta Ct}$ method. The primers used are as follows: HIF-1-$\alpha$, 5′-AGCGACGTGGCTATTGTGAAG-3′(forward) and 5′-GCCATCATTCTTGAGGAGGAAGT-3′ (reverse); 18S, 5′-GAACGAGACTCTGGCATGCTA-3′ (forward) and 5′-CACGCTGAGCCAGTCAGTGTA-3′ (reverse); VEGFA, 5′-AGGGCAGAATCATCACGAAGT-3′ (forward) and 5′-AGGGTCTCGATTGGATGGCA-3′ (reverse); IL-1$\alpha$, 5′-TGTATGTGACTGCCCAAGATGAAG-3′ (forward) and reverse: 5′-AGAGGAGGTTGGTCTCACTACC-3′ (reverse); IL-1 $\beta$, 5′-CCAGCTTCAAATCTCACAGCAG-3′ (forward) and reverse: 5′-CTTCTTTGGGTATTGCTTGGGATC-3′ (reverse); IL-6, 5′-CCTCCAGAACAGATTTGAGAGTAGT-3′ (forward) and 5′-GGGTCAGGGGTGGTTATTGC-3′ (reverse); IL-8, 5′-AAGACATACTCCAAACCTTTCCACC-3′ (forward) and 5′-CTTCAAAAACTTCTCCACAACCCTC-3′ (reverse); IL-10, 5′-AACCTGCCTAACATGCTTCGA-3′ (forward) and 5′-CTCATGGCTTTGTAGATGCCT-3′ (reverse); TNF-$\alpha$, 5′-CCCGCATCCCAGGACCTCTCT-3′ (forward) and 5′-CGGGGGACTGGCGA-3′ (reverse).

## Xenograft experiments

Experiments were performed under a project license (No. SHCH-IACUC-2020-XMSB-36) granted by the Animal Ethics and Welfare Committee of Shanghai Children's Hospital, in compliance with National Institutes of Health (NIH) Guide for the Care and Use of Animals. Five-week-old BALB/c male nude mice were purchased from Shanghai Laboratory Animal Center, China. The mice were housed in a specific-pathogen-free (SPF) environment with temperature set between $22 \pm 2°C$ under 12 h light/dark cycle and relative humidity of $60 \pm 10\%$. All the mice were given an unlimited supply of sterile food and water.

HemECs ($5 \times 10^6$) were inoculated into the flanks of the male BALB/c nude mice. Tumors were allowed to grow for 4 days. The initial tumor volume measurement was

made using calipers. The mice were randomly divided into either the control, $\beta$-elemene, propranolol, or $\beta$-elemene + propranolol groups (four mice in each group). Mice were then treated daily with peritumoral injections of PBS (control group), $\beta$-elemene (75 mg/kg), propranolol (50 mg/kg), or $\beta$-elemene + propranolol. Tumors were measured with a caliper every 7 days, and the tumor volume was calculated using the formula V = length × width$^2$/2. After 30 days of treatment, mice were euthanized using carbon dioxide, and the tumors were dissected and measured. The data are presented as mean ± SD.

### Immunofluorescent staining

Tumors were embedded into Tissue-Tek optimum cutting temperature freezing media, cut into 5 μm cross sections, and fixed with 95% ethanol for 10 min. The sections were incubated with a CD31 primary antibody at 4 °C overnight after blocking. The tissue sections were then rinsed in phosphate buffered saline (PBS). The next day, the samples were incubated with a mixture of the corresponding secondary antibodies and imaged using an optical microscope. Microvessel density (MVD) was analyzed and measured using CD31 staining.

### Statistical analysis

All results were repeated independently a minimum of three times and in triplicate. GraphPad Prism 5.0 was used for analysis. The differences among groups were measured using A two-tailed unpaired $t$-test and ANOVA. A $p$-value <0.05 was considered statistically significant.

## RESULTS

### Cytotoxic effect of propranolol and $\beta$-elemene on HemECs

The cytotoxic effects of $\beta$-elemene and propranolol were detected using an LDH assay. HemECs were incubated with different concentrations of $\beta$-elemene (25, 50, 75, 100, and 125 μg/ml) or propranolol (25, 50, 75, and 100 μM) for 24 h. The results showed no significant difference in the cytotoxic effect of $\beta$-elemene at concentrations lower than or equal to 75 μg/ml. Furthermore, there were no apparent cytotoxic effects of propranolol on HemECs at concentrations lower than or equal to 25 μM (Figs. 1A–1B). Cellular proliferation was measured *via* a CCK-8 method, as shown in Figs. 1C–1D. The proliferation of HemECs was inhibited at concentrations of $\beta$-elemene higher than or equal to 75 μg/ml, as well as at concentrations of propranolol that were higher than or equal to 50 μM. The results revealed that $\beta$-elemene was non-cytotoxic at the concentration of 75 μg/ml and produced inhibitory effects. Moreover, propranolol was cytotoxic at concentrations of 50 μM and produced inhibitory effects. To further study the cytotoxic effects of propranolol (50 μM, and 75 μM) and $\beta$-elemene (75 μg/ml), Calcein-AM/PI double staining was used. It was found that compared with the control, $\beta$-elemene had no apparent cytotoxic effects and propranolol at concentrations of 50 μM and 75μM was cytotoxic. Furthermore, by improving the propranolol concentration its cytotoxicity gradually increased. Combinatorial treatment using $\beta$-elemene and propranolol did not increase cytotoxicity compared to monotherapy with propranolol. The results suggest that

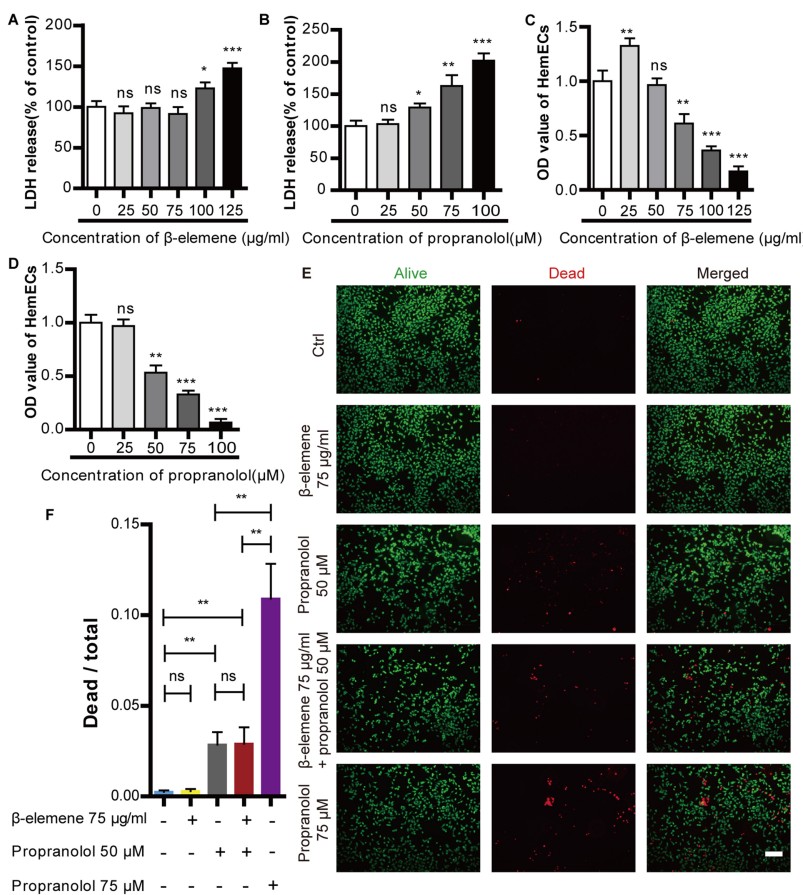

**Figure 1 Cytotoxic effect of propranolol and $\beta$-elemene on HemECs.** (A) A concentrations of $\beta$-elemene lower than or equal to 75 $\mu$g/ml did not produce a cytotoxic effect on HemECs. (B) There was no apparent effect in the cytotoxicity of propranolol on HemECs at concentrations lower than or equal to 25 $\mu$M. (C) Compared with the control group, proliferation of the HemECs decreased at concentrations of $\beta$-elemene more than 75 $\mu$g/ml, the data was evaluated *via* a cell counting kit-8 assay. (D) Compared to the control group, proliferation of the HemECs decreased at concentrations of propranolol more than 25 $\mu$M, the data was evaluated *via* a Cell counting kit-8 assay. (E) Representative photomicrographs for Calcein-AM/PI double staining of the HemECs in five different groups: control, $\beta$-elemene (75 $\mu$g/ml), propranolol (50 $\mu$M), combinatorial treatment with $\beta$-elemene (75 $\mu$g/ml) and propranolol (50 $\mu$M), and the propranolol groups (75 $\mu$M) (scale bar = 100 $\mu$m). (F) Quantification of Calcein-AM/PI double staining. The experiment was repeated in triplicate and results are presented as mean $\pm$ SD; *, $p < 0.05$; **, $p < 0.01$; ***, $p < 0.001$; *ns*, not significant.

combinatorial therapy with $\beta$-elemene and propranolol did not increase the cytotoxic effects of HemECs (Figs. 1E–1F). Therefore, 75 $\mu$g/ml $\beta$-elemene and 50 $\mu$M propranolol were used in subsequent experiments.

## Combinatorial treatment using $\beta$-elemene and propranolol inhibited the migration, proliferation, and tube formation of HemECs without increasing cytotoxic side effects

Cellular migration was analyzed *via* a scratch test, cell proliferation was detected using the CCK-8 method, and tube formation was detected *via* a tube formation assay in order to

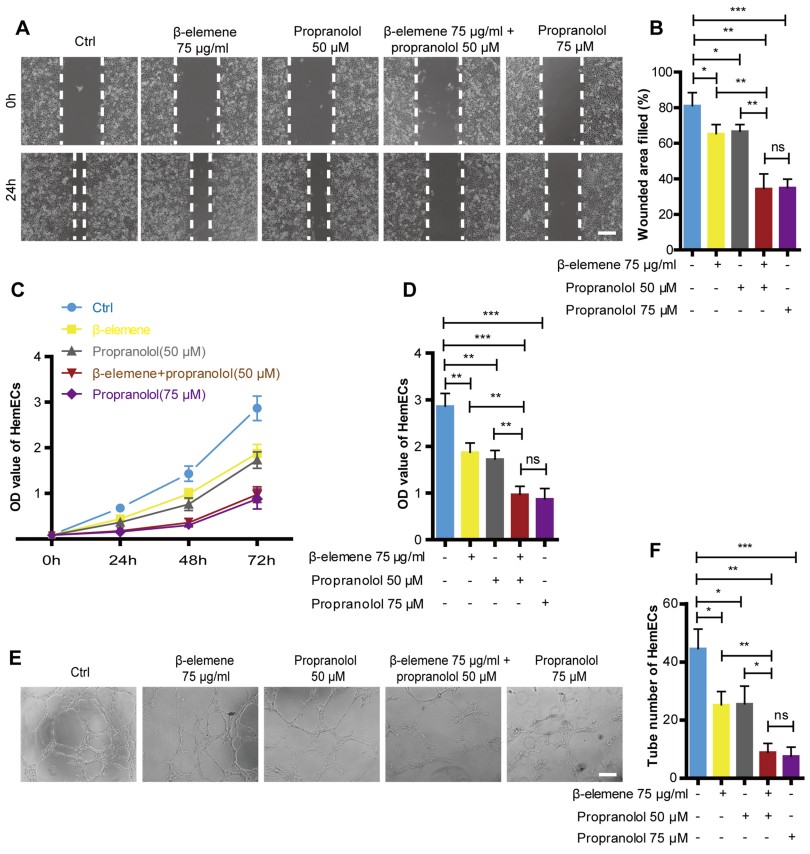

**Figure 2 Combinatorial treatment of β-elemene and propranolol inhibited the migration and proliferation and tube formation of HemECs.** (A) Representative photomicrographs of the HemECs scratch test of five different groups: control, β-elemene (75 μg/ml), propranolol (50 μM), and combinatorial treatment with β-elemene (75 μg/ml) and propranolol (50 μM), and propranolol groups (75 μM) (scale bar = 250 μm). (B) Quantitative analyses of the relative migration rate. (C) HemECs treated with β-elemene and propranolol for 24 h, 48 h, and 72 h. The proliferation of the HemECs were evaluated using the cell counting kit-8 assay. (D) Quantification of proliferation in HemECs treated with β-elemene and propranolol for 72 h. (E, F) Representative photomicrographs of the tube formation assay in HemECs within five different groups. The corresponding quantitative analysis of the tube number of the capillary tubes formed by the HemECs (scale bar = 200 μm). The experiment was repeated in triplicate and results are presented as mean ± SD; *, $p < 0.05$; **, $p < 0.01$; ***, $p < 0.001$; *ns*, not significant.

investigate the combinatorial effects of β-elemene and propranolol on the migration and proliferation of HemECs. As shown in Fig. 2, compared with the single treatment group, combinatorial therapy using β-elemene (75 μg/ml) and propranolol (50 μM) significantly inhibited the migration, proliferation, and tube formation of HemECs. Additionally, the combinatorial treatment had the same inhibitory effects as the high-concentration propranolol treatment group (75 μM). Moreover, previous results showed that compared to the high-concentration propranolol group (75 μM), that combinatorial treatment with β-elemene (75 μg/ml) and propranolol (50 μM) had lower cytotoxicity. The results suggest that β-elemene combined with propranolol enhanced the inhibitory effect of HemECs without increasing the cytotoxic side effects.

### HIF-1-α/VEGFA signaling pathway was activiated in hemangioma

To reveal how the HIF-1-α/VEGFA signaling pathway affects hemangioma proliferation, migration, and angiogenesis, We tested the expression of HIF-1-α, phospho-Akt, Akt, phospho-Erk, Erk, and VEGFA on protein levels in HUVECs and HemECs using Western Blot. The results showed that expression of HIF-1-α, phospho-Akt, phospho-Erk and VEGFA were higher than that of the HUVECs (Figs. 3B–3C). The RT-PCR results were consistent with that of the western blotting (Fig. 3A). Functional analysis confirmed that the ability of proliferation, migration, and tube formation in HemECs were more significant than that in the HUVECs (Figs. 3D–3H). Overall, our data demonstrated that HemECs compared with HUVECs, expressed higher levels of HIF-1-α and VEGFA, and exhibited a stronger growth potential. This conclusion helped us to further evaluate the effects of the combination of propranolol on HA.

### Combinatorial treatment of β-elemene and propranolol inhibited the expression of HIF-1-α and VEGFA in HemECs

As previously reported, the HIF-1-α/VEGFA signaling pathway plays an important role during proliferation, migration, and angiogenesis. To evaluate the combinatorial treatment's underlying mechanism, we divided the cells into specific groups; the control, β-elemene, propranolol, and combinatorial treatment groups and then exposed each group to hypoxia for 24 hours. Our study showed that compared with the control group, β-elemene and propranolol inhibited the expression of HIF-1-α, VEGFA, p-Akt/Akt, and p-Erk/Erk. However, the combinatorial treatment using β-elemene and propranolol produced significantly more inhibitory effects (Figs. 4C–4D). The RT-PCR results were consistent with that of the western blotting (Figs. 4A–4B), suggesting that combinatorial treatment synergistically down-regulates the HIF-1-α/VEGFA signaling pathway.

### Combinatorial treatment of β-elemene and propranolol inhibited the cytokines in HemECs

It has been widely accepted that chronic inflammation increases tumorigenesis, development, and metastasis by creating an immunosuppressive microenvironment. As an inducer of tumors, inflammation could increase the risk of cancer by providing bioactive molecules, like cytokines, from cells infiltrating the tumor microenvironment. Therefore, inhibition of inflammation appears to be a promising therapy for anti-tumor treatment.

Our study showed that compared with the HUVECs, the expression of IL-6, IL-8 and TNF-α were increased in HemECs (Fig. 5A). We further evaluated the expression of cytokines in combinatorial treatment of β-elemene and propranolol. The data showed that the expression of cytokines were significantly inhibited in drug treatment groups compared with control, and combinatorial treatment of β-elemene and propranolol showed the better inhibitory effects than single treatment with β-elemene or propranolol (Fig. 5B).

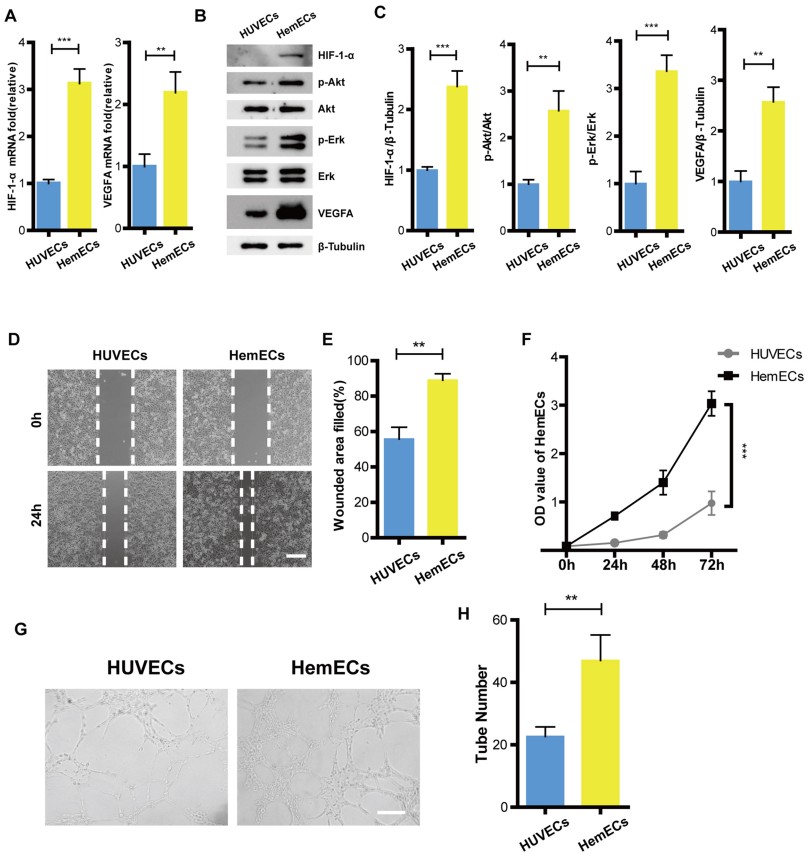

**Figure 3** **HIF-1-α/VEGFA signaling pathway was acitivied in hemangioma.** (A) The expression of HIF-1-α and VEGFA observed by RT-PCR. (B) The expression of HIF-1-α, phospho-Akt, Akt, phospho-Erk, Erk, and VEGFA observed by western blotting. (C) Quantified protein expression levels of HIF-1-α, VEGFA, p-Akt/Akt, and p-Erk/Erk. (D) Representative photomicrographs of the HUVECs and HemECs scratch test (scale bar = 250 μm). (E) Quantitative analyses of the relative migration rate. (F) Compared with HUVECs, proliferation of the HemECs were increased, data was evaluated *via* a cell counting kit-8 assay. (G) Representative photomicrographs of the tube formation assay in HUVECs and HemECs (scale bar = 200 μm). (H) Quantitative analysis of the tube number of the capillary tubes . The experiment was repeated in triplicate and results are presented as mean ± SD; *, $p < 0.05$; **, $p < 0.01$; ***, $p < 0.001$; *ns*, not significant.

## Combinatorial treatment of *β*-elemene and propranolol inhibited the growth and angiogenesis of hemangioma

The suppressive effects of the combinatorial treatment of *β*-elemene and propranolol have been confirmed in HemECs. These effects were further validated in a mice xenograft tumor model. After 30 days of treatment using 75 mg/kg *β*-elemene and 50 mg/kg propranolol daily in nude mice, the tumor growth curve revealed that the tumor volume of the combinatorial treatment group was significantly smaller than that of the control, propranolol, and *β*-elemene monotherapy treatment groups (Figs. 6B–6C). Separated tumors of the HemEC cell xenografts and tumor weight showed consistent results (Fig. 6A and 6D). The CD31 antibody (red) was used as a microcapillary marker during immunofluorescence. As shown in Figs. 6E–6F, compared with the control, propranolol,

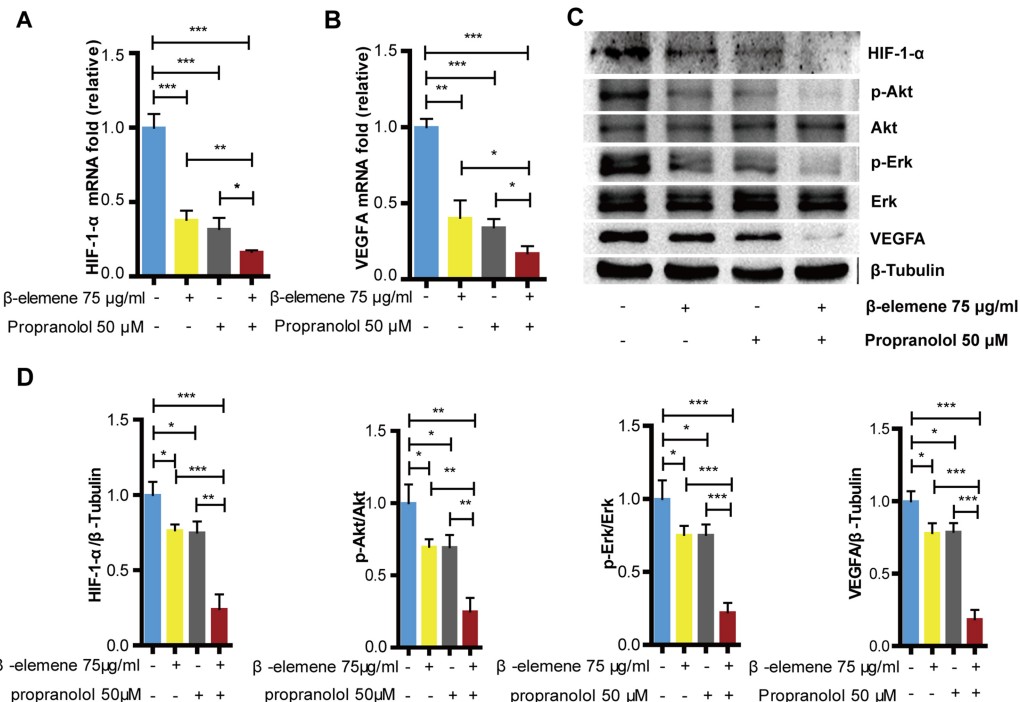

**Figure 4 Combinatorial treatment of $\beta$-elemene and propranolol inhibits the expression of HIF-1-$\alpha$ and VEGFA in HemECs.** HemECs treated with $\beta$-elemene and propranolol for 24 h, (A and B) The expression of HIF-1-$\alpha$, and VEGFA observed by RT-PCR. (C) The expression of HIF-1-$\alpha$, VEGFA, p-Akt/Akt, and p-Erk/Erk observed by western blotting. (D) Quantified protein expression levels of HIF-1-$\alpha$, VEGFA, p-Akt/Akt, and p-Erk/Erk. The experiment was repeated in triplicate and results are presented as mean $\pm$ SD; $^*$, $p < 0.05$; $^{**}$, $p < 0.01$; $^{***}$, $p < 0.001$; $ns$, not significant.

or $\beta$-elemene monotherapy treatment groups, angiogenesis decreased significantly in the combinatorial treatment group. The results suggested that combinatorial treatment with $\beta$-elemene and propranolol significantly inhibited the growth and angiogenesis of hemangioma.

## DISCUSSION

Hemangioma is one of the most common benign endothelial cell tumors characterized by excessive proliferation of immature blood vessel formation (*Boscolo & Bischoff, 2009*). $\beta$-Adrenoceptor blockers are among the most widely used drugs in clinical practice to treat cardiovascular diseases such as hypertension, heart failure, and cardiac arrhythmias, as well as exert cardio protective effects (*Agesen et al., 2019*). In 2008, $\beta$-Adrenoceptor blockers were first shown to treat infantile hemangioma, and then were used worldwide as a first-line therapy for hemangioma due to their increased efficacy (*Leaute-Labreze et al., 2008*; *Price et al., 2011*; *Tran et al., 2016*). Various studies have been performed that have investigated the mechanisms by which propranolol induces the involution of hemangioma. However, detrimental side effects caused by propranolol include: light-headedness, drowsiness, nausea, cold extremities, fatigue, bradycardia, bronchospasm, indigestion, and diarrhea

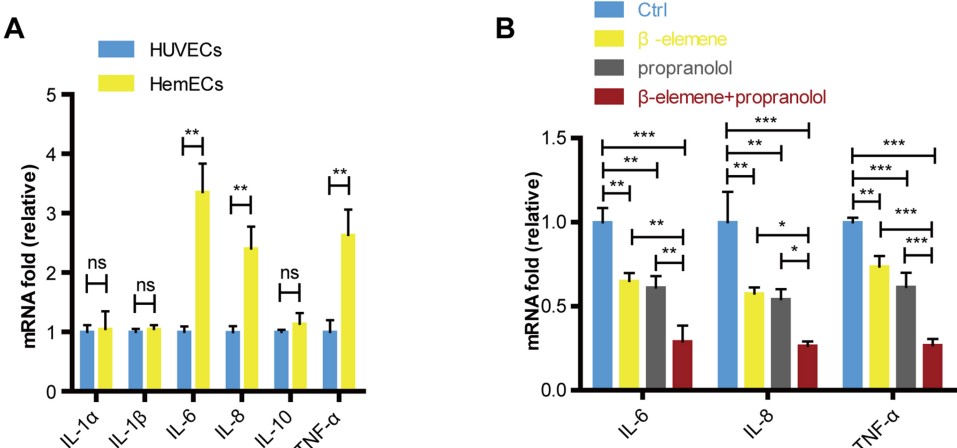

**Figure 5** **Combinatorial treatment of β-elemene and propranolol inhibited the cytokines in HemECs.**
(A) The expression of IL-1 α, IL-1 β, IL-6, IL-8, IL-10, TNF- α in HUVECs and HemECs observed by
RT-PCR. (B) HemECs treated with β-elemene and propranolol for 24 h, the expression of IL-6, IL-8 and
TNF- α observed by RT-PCR. The experiment was repeated in triplicate and results are presented as mean
± SD; *, $p < 0.05$; **, $p < 0.01$; ***, $p < 0.001$; ns, not significant.

(*Leaute-Labreze et al., 2016*; *Leaute-Labreze, Harper & Hoeger, 2017*; *Seebauer et al., 2022*).
Therefore, the clinical application of β-Adrenoceptor blockers still has certain limitations.
The present study found that propranolol is cytotoxic at concentrations (≥ 50 μM)
and produces functional effects. Increasing the inhibitory effects of propranolol without
increasing cytotoxicity is a key issue that needs to be solved. In recent years, combinatorial
therapies have become a popular topic of research. Therefore, we investigated whether it is
possible to find a potential drug combination in order to improve the efficacy and reduce
the side effects of propranolol.

β-elemene is a bioactive compound isolated from the traditional Chinese medicinal
herb Rhizoma zedoariae. It exerts a wide range of antitumor activities and low toxicity
(*Chen et al., 2020*). Previous studies have shown that β-elemene inhibits melanoma
growth by suppressing VEGF (*Chen et al., 2011*). β-elemene inhibits thyroid carcinoma
proliferation and angiogenesis (*Zhao et al., 2020*), suggesting that β-elemene is a potential
anti-angiogenic drug. Wang et al. tested β-elemene for its adverse side effects at different
concentrations and the results showed that β-elemene had no obvious toxic effects on
neurons at concentrations lower than 40 μg/ml and was toxic at equal to or higher than
80 μg/ml (*Wang et al., 2018*). Our previous research has shown that β-elemene has no
apparent toxic effects on hemangioma cells at concentrations lower than or equal to
80 μg/ml (*Wang et al., 2021*). In this experiment, we choose differents concentrations
of β-elemene, and found that β-elemene was non-cytotoxic at a concentration of 75
μg/ml and produced inhibitory effects. However, the inhibitory effect of β-elemene at
this concentration on hemangioma is still limited. Increased concentrations of β-elemene
produced greater inhibitory effects, as well as increased cytotoxic side effects. Therefore,
we came up with the idea of a combinatorial treatment approach using β-elemene and

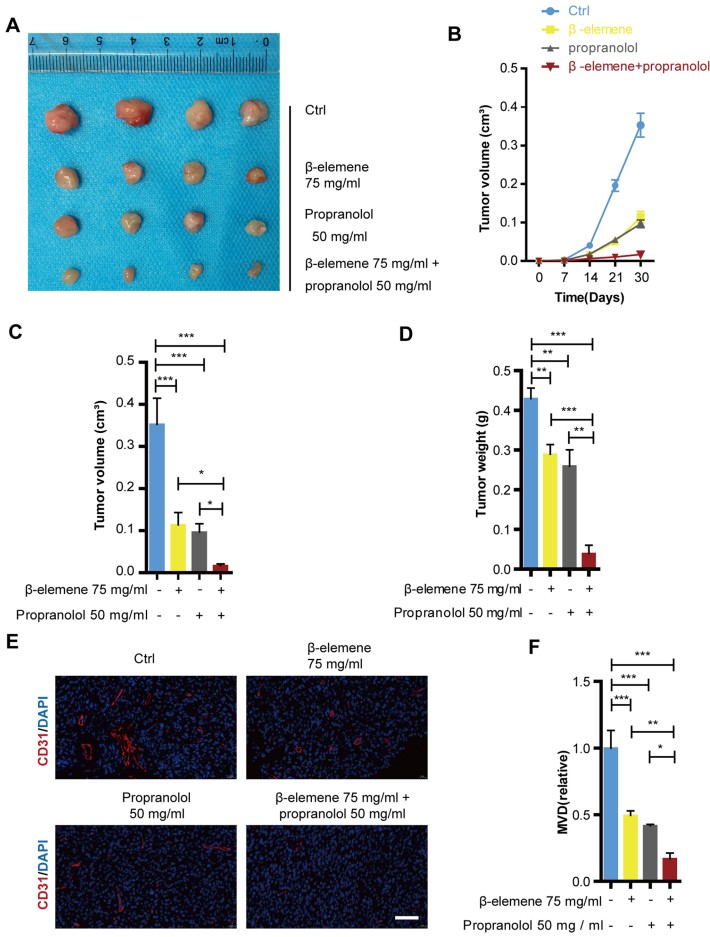

**Figure 6** **Combinatorial treatment of $\beta$-elemene and propranolol inhibited growth and angiogenesis of hemangioma.** The HemECs were injected subcutaneously into the left flank of the nude mice. After 30 days of $\beta$-elemene (75 mg/kg d) and propranolol (50 mg/kg d) treatment the primary hemangioma tumor was carefully removed from the left flank for analysis. (A) Hemangioma images (scale bar 1 cm). (B) The size of the hemangioma before resection. (C) Statistical analysis of tumor volume after tumors separated. (D) Statistical analysis of tumor weight. (E) Immunofluorescence used to detect the expression of CD31 on hemangioma (scale bar 200 $\mu$m). (F) Quantification of microvessel density (MVD). The experiment was repeated in triplicate and results are presented as mean $\pm$ SD; *, $p < 0.05$; **, $p < 0.01$; ***, $p < 0.001$; *ns*, not significant.

propranolol. In this research, we found that the combinatorial treatment of $\beta$-elemene and propranolol did not increase the cytotoxic effects of HemECs and exerted significantly more inhibitory effects, suggesting that $\beta$-elemene is an excellent potential combination drug addition.

Angiogenesis, the physiological process of new blood vessel formation from pre-existing capillaries, is critical for expanding and remodeling the vascular network that involves the proliferation and migration of endothelial cells (*Lee et al., 2021*; *Yang et al., 2022*). The process's imbalance leads to the pathogenesis of numerous diseases, such as tumor growth and metastasis. The growth of tissues requires blood vessels in order to transport

nutrients by transcriptionally activating several angiogenic genes and their receptors. Every step of angiogenesis is supported by hypoxia and HIF-1 (*Conway, Collen & Carmeliet, 2001*). When tissues are exposed to hypoxia, various cell types in the body respond by up-regulating hypoxia-inducible factor-1 alpha; therefore, inducing the expression of many angiogenic genes, including vascular endothelial growth factors (VEGF). Additionally, VEGFA is thought to promote angiogenesis by stimulating endothelial cell proliferation and migration (*Yamamoto et al., 2020*). VEGF is a major transcriptional target of HIF-1 and promotes angiogenesis (*Xu et al., 2022*). Furthermore, hypoxia and HIF-1 are often dependent on VEGF levels (*Liu et al., 1995*). HIF-1 upregulates VEGF directly under hypoxic conditions in order to promote the recruitment of endothelial progenitor cells and induce their differentiation into endothelial cells.

Hemangioma is characterized by excessive proliferation of vascular endothelial cells and immature blood vessel formation (*Boscolo, Mulliken & Bischoff, 2013*; *Wu et al., 2021a*). Previous research has shown that the expression of HIF-1-$\alpha$ is increased in hemangioma; thus, over-expression of HIF-1-$\alpha$ promotes angiogenesis suggesting that the angiogenesis of hemangioma is driven by hypoxia (*Chen et al., 2017*). Numerous reports have confirmed that excessive VEGF expression in HA tissue parallels the proliferating phase of its growth. Additionally, the expression of VEGF rapidly decreases as many angiogenesis inhibitors become prominent in the involuting phase (*Ji et al., 2014*). Chim et al. demonstrated that propranolol exerts its suppressive effects on hemangioma through the HIF-1-$\alpha$/VEGF-A angiogenesis axis (*Chim et al., 2012*). Propranolol-induced cell growth arrest was also abrogated by HIF-1-$\alpha$ overexpression. These studies suggested that the VEGFA axis is crucial for the growth and angiogenesis of hemangioma. Consistent with previous studies, our data showed that HIF-1-$\alpha$ and VEGFA expression increases in HemECs. Furthermore, in the previous research, we were able to show that expression of HIF-1-$\alpha$ and VEGFA was decreased within the combinatorial group *versus* the single-treated group. Xenograft experiments showed that combinatorial treatment using $\beta$-elemene and propranolol significantly inhibited hemangioma growth and angiogenesis.

## CONCLUSION

In summary, our data found that the combinatorial treatment using propranolol and $\beta$-elemene not only did not increase the toxic side effects of the HemECs, but significantly enhanced the inhibitory effects of proliferation, migration, and tube formation. The results of the HemECs xenograft experiment were consistent with the functional experiments *in vitro*. We further revealed the mechanism of action of the combinatorial treatment: $\beta$-elemene in combination with propranolol inhibits proliferation, migration, and angiogenesis of hemangioma by synergistically down-regulating the HIF-1-$\alpha$/VEGFA signaling pathway (Fig. 7). These results suggest that combinatorial treatment with $\beta$-elemene and propranolol is a novel strategy for treating hemangioma, or diseases where the pathogenesis is based on angiogenesis.
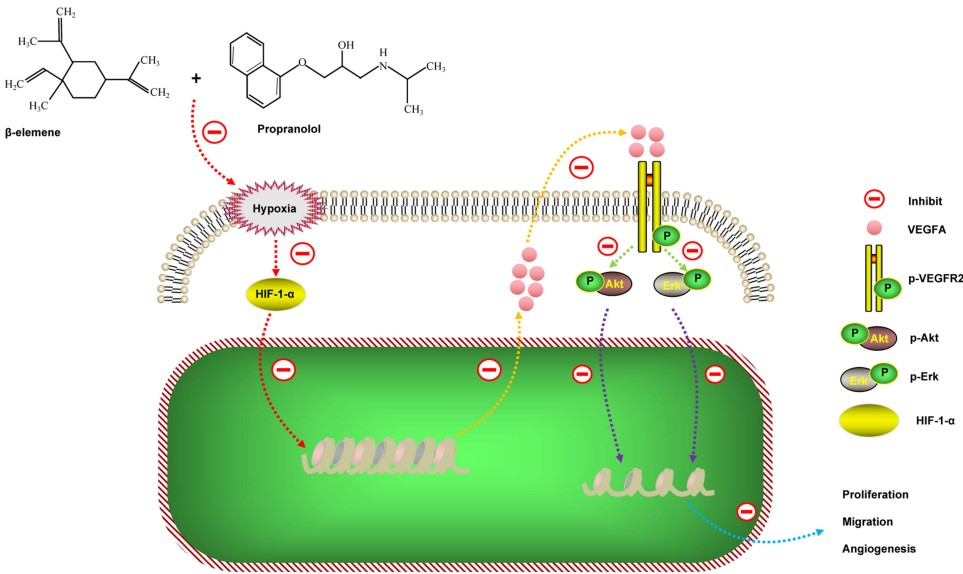

**Figure 7** Schematic diagram of the possible mechanisms of action of the combinatorial treatment using $\beta$-elemene and propranolol on hemangioma. $\beta$-elemene in combination with propranolol inhibits the expression of VEGFA by synergistically down-regulating HIF-1-$\alpha$ expression. Decreased VEGFA expression inhibits VEGFR2 activation, which in turn affects the phosphorylation of downstream signaling molecules, including Akt and Erk; therefore, ultimately inhibiting hemangioma growth and angiogenesis.

# ACKNOWLEDGEMENTS

The authors thank AiMi Academic Services for the English language editing.

## Funding
The authors received no funding for this work.

## Competing Interests
The authors declare there are no competing interests.

## Author Contributions
- Zhenyu Wang performed the experiments, authored or reviewed drafts of the article, and approved the final draft.
- Yinxian Chen conceived and designed the experiments, authored or reviewed drafts of the article, and approved the final draft.
- Lin Yang analyzed the data, prepared figures and/or tables, and approved the final draft.
- Dunbiao Yao analyzed the data, prepared figures and/or tables, and approved the final draft.
- Yang Shen analyzed the data, prepared figures and/or tables, authored or reviewed drafts of the article, and approved the final draft.

## Animal Ethics

The following information was supplied relating to ethical approvals (*i.e.*, approving body and any reference numbers):

The authors are accountable for all aspects of the work in ensuring that questions related to the accuracy or integrity of any part of the work are appropriately investigated and resolved. Experiments were performed under a project license (No. SHCH-IACUC-2020-XMSB-36) granted by Animal Ethics and Welfare Committee of Shanghai Children's Hospital, in compliance with National Institutes of Health (NIH) Guide for the Care and Use of Animals.

## Data Availability

The data is available at figshare: Wang, Zhenyu (2023). Combinative effects of *β*-elemene and propranolol on the proliferation, migration, and angiogenesis of hemangioma. figshare. Dataset. https://doi.org/10.6084/m9.figshare.22722949.v1.

## Supplemental Information

Supplemental information for this article can be found online at http://dx.doi.org/10.7717/peerj.15643#supplemental-information.

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
