# Peer review of "Combinative effects of β-elemene and propranolol on the proliferation, migration, and angiogenesis of hemangioma"

_PeerJ, doi:10.7717/peerj.15643_

## Round 0.1 · original submission · Minor Revisions

Please carefully read the comments and suggestions from the reviewers and provide your point-to-point responses.

·

Basic reporting

1. English needs further polishing.
2. Please check some formatting issues carefully.
3. Scale bars should be added to Figure 3c, 3f.

Experimental design

1. Why are HUVECs and HemECs compared? What's the difference between the two cells?
2. Why is it associated with the HIF-1-³/VEGFA signaling pathway in Figure3? Is there any good reason?

Validity of the findings

No comment

Reviewer 2 ·

Basic reporting

This article takes the hemangioma-derived endothelial cells (HemECs) as the target to discuss the specific mechanism of the combined application of β‐elemene and propranolol in the treatment of hemangioma (HA), which is of great significance for the clinical treatment of HA. However, there are still some problems in this manuscript that need to be explained by the author:
1.Why there is no mRNA quantification for HIF-1-α and VEGFA in Figure 3.
2.The scale bar should be displayed in the images of the scratch test and tube formation assay in Figure 3.
3.The punctuation is wrong on page 6, line 133.
4.The ‘HUVEC’ in result 2 (Line 232) and the ‘HUVECs’ in the figure 3 need to be unified in writing.
5.“The RT-PCR results are consistent with ~” on page 9, Line 249: “The RT-PCR results were consistent with ~”.
6.The punctuation is wrong on page 12, line 347. Please correct.
7.“The results of the HemEC cell xenograft experiment are consistent with ~” on page 12, Line 347: “The results of the HemEC cells xenograft experiment were consistent with ~”.
8.In the part of results, how did the author determine the concentration gradient of β-elemene and propranolol ?

Experimental design

no comment

Validity of the findings

no comment

Reviewer 3 ·

Basic reporting

The manuscript by Wang et al. focuses on the effects of the combined action of β-elemene and propranolol on the proliferation, migration and angiogenesis of hemangioma, their results indicated that the combinatorial treatment using propranolol and β-elemene not only did not increase the toxic side effects of the HemEC cells, but significantly enhanced the inhibitory effects of proliferation, migration, and tube formation, which is interesting, providing new ideas and strategies for the treatment of hemangioma. Despite some small typos and mistakes, the manuscript was well-written using professional and easy-understandable language,the overall writing is clear. However, there are some concerns which should be addressed as follows:
1. There are many language and punctuation mark errors in the manuscript (such as line 38, 48,49, 57, et al.), please check.
2. In line 101, the abbreviation of Hemangioma-derived endothelial cells has been mentioned above, which should not appear again, please check.
3. There are some symbol fonts errors in the text, such as ‘±’ in line 171,which should be changed to ‘±’ (Times New Rmoman), and ‘≥’ in line 294,which should be changed to ‘≥’ (Times New Rmoman), et al., please check.
4.The scale bar is not indicated in the Figure 3, please add.
5. In line 280-281, reference is too old, please update.
6. In line 293, β-blockers should be changed as “β-Adrenoceptor blockers”, please check.

Experimental design

The experimental design was straightforward and easy to follow. The results can support the relevant conclusions. However, it should be noted that
in the method section, how is the dose of β-elemene determined?

Validity of the findings

The raw data was supplied.The results of the manuscript is novel and can serve as a reference for further research on angiogenesis.

Additional comments

In this manuscript, HIF-1-α is selected as the focus of the research, is VEGFA signaling pathway related to the basis of previous research, or why such a signaling pathway is selected for research?

---

## Round 0.2 · accepted · Accept

The authors have addressed reviewers' concerns and the paper may be acceptable for publication.

·

Basic reporting

No comment.

Experimental design

No comment.

Validity of the findings

No comment.

Additional comments

No comment.

Reviewer 2 ·

Basic reporting

no comment

Experimental design

no comment

Validity of the findings

no comment

Reviewer 3 ·

Basic reporting

no comment

Experimental design

no comment

Validity of the findings

no comment

Additional comments

Authors addressed my concerns. I have no further comments.